# Pectenovarin, A New Ovarian Carotenoprotein from Japanese Scallop *Mizuhopecten yessoensis*

**DOI:** 10.3390/molecules25133042

**Published:** 2020-07-03

**Authors:** Satoko Matsunaga, Hiroki Ikeda, Ryuichi Sakai

**Affiliations:** 1Department of Material and Environmental Engineering, National Institute of Technology HAKODATE College, 14-1 Tokura-cho, Hakodate, Hokkaido 042-8501, Japan; 2Faculty and Graduate School of Fisheries Sciences, Hokkaido University, 3-1-1 Minato-cho, Hakodate 041-8611, Japan; mhpta2000@eis.hokudai.ac.jp (H.I.); ryu.sakai@fish.hokudai.ac.jp (R.S.)

**Keywords:** carotenoproteins, carotenoids, scallop ovary, Japanese scallop

## Abstract

The scallop *Mizuhopecten yessoensis* accumulates carotenoids in the ovary during the maturation stage. Its conspicuous pink color implies the presence of carotenoprotein. However, the carotenoprotein from the scallop ovary has never been isolated and characterized, probably due to its instability and complexity. Here, we developed an extraction and isolation procedure for the carotenoprotein by employing a basic buffer containing potassium bromide to facilitate its efficient extraction from the ovary, and we succeeded in obtaining the carotenoprotein, termed pectenovarin. The carotenoid composition of the pectenovarin was similar to that of the ovary. The N-terminal and internal amino acid sequences of pectenovarin showed a high similarity to those of vitellogenin, the precursor of egg yolk protein under analysis.

## 1. Introduction

Carotenoids are one of the essential classes of metabolites in a wide variety of living organisms. More than 750 different carotenoids have been reported to date [1]. It is known that carotenoids bind to proteins, termed carotenoproteins, to stabilize themselves against light or heat or to manipulate their physical properties, including solubility and color. Crustacyanin, one of the most representative carotenoproteins, binds to a red keto carotenoid, astaxanthin, but exhibits conspicuous blue color because of the bathochromic shift in the visible light region, due to the interaction between the carotenoids and the apoprotein. The structural evidence for specific binding between astaxanthin and the protein was observed by X-ray crystallography [2]. Although, the blue color of crustacyanin is stable in an ambient milieu, this conspicuous color turns reddish-orange by heating or immersion in organic solvents. The molecular basis of the blue shift is an intriguing subject for biophysical research [2,3,4].

The Japanese scallop *Mizuhopecten yessoensis* accumulates carotenoids in the ovary during the gonadal maturation process. These carotenoids are mainly pectenolone, but pectenoxanthin (alloxanthin), diatoxanthin, 3,4,3′-trihydroxy-7′-8′-didehydro-β-carotene, and astaxanthin are known to be minor constituents [5]. These molecules are accumulated mostly in the ovary, but not in other parts of the body, including the viscera, testes in males, and adductor muscle, and are thought to bear functions such as anti-oxidation and light protection [6]. These carotenoids originate from food microalgae and are likely subject to structural transformation, then accumulated in the form of carotenoproteins in the ovary. However, their actual functions in the reproduction of shellfish are not precisely known. Although there have been several reports dealing with the characterization and utilization of carotenoids in scallops [5,6,7,8], isolation and characterization of the carotenoprotein in the ovary have never been reported to the best of our knowledge. We, therefore, conducted isolation and characterization of the carotenoprotein from the ovary of *M. yessoensis.* Here, the extraction, purification, and N-terminal and internal amino acid sequences of the new carotenoprotein, named pectenovarin, are described.

## 2. Results

### 2.1. Extraction of the Colored Protein from the Scallop Ovary

We first used water to extract the pink color of the scallop ovary. However, the extract became cloudy, and a clear solution could not be obtained even after centrifugation. Therefore, we optimized the conditions to extract the pink material stably into an aqueous extract. First, the ovarian tissue was homogenized with a basic buffer, 0.05 M Tris-HCl, pH 8.0. Although the centrifuged homogenate gave pink-colored supernatant, it was still cloudy, and the color remained mostly in the precipitate. We noticed that extractability varied from individual to individual, and in some specimens, the same procedure failed to extract colored material into the solution. When an acidic buffer (pH 3.5) was added to the above basic extract, the pink material was completely precipitated and never re-solubilized. To the contrary, the addition of 20 mM NaCl to the above extract afforded a clear supernatant. These results indicate that the pink component is not extractable under acidic conditions, but solubilizes under basic conditions. Importantly, high ionic strength was required to obtain a “clear” extract. Although ammonium sulfate (AS) precipitation yielded the recovery of deep pink pellets at about 30% AS, the color faded during storage at 4 °C for a few days. These results together revealed that the pink-colored protein is difficult to extract and is unstable. We further screened the extraction conditions, focusing on pH and ionic requirements. Extraction under neutral conditions at pH 7.2 resulted in complete precipitation of the pink material and a cloudy, pale orange supernatant, as expected. Re-solubilization of the pellet with basic buffer (Tris-HCl pH 8.0, with 150 mM NaCl) further extracted the colored material to the supernatant to some extent, but color still remained in the pellet. Therefore, we employed KBr-containing buffer for extraction according to a procedure of lipoprotein extraction [9], since the pink component was likely to have a lipoprotein-like nature. We extracted a new ovary using a basic buffer containing KBr, and we succeeded in extracting the pink color in the supernatant efficiently. A deep pink pellet was obtained by ammonium sulfate precipitation (20%). These experiments established conditions to extract the pink color efficiently from the scallop ovary, requiring three factors: basic conditions and the presence of NaCl and KBr.

### 2.2. Isolation of Pectenovarin

In order to isolate colored protein, which is named “pectenovarin” by us, from the pink-colored extract, gel filtration using Sephacryl S-300 (optimized for M*r* 10^4^–10^6^ Da) was conducted. However, the colored protein was eluted at the solvent front without separation, suggesting that the intact protein was larger than 10^6^ Da. Anion exchange chromatography (DE52) failed due to residual salt that was essential to stabilize the protein. A hydrophobic column absorbed the protein but eluted over large volumes and thus did not result in reasonable separation. We directly applied the KBr extract to gel filtration high-pressure liquid chromatography (HPLC) (BioSec-5), using the above buffer without KBr as eluent, and separated into nine fractions (Figure 1A,B); of these, fraction 4 had noticeable pink color, while fraction 5 had very weak pink color. The photodiode array data for fraction 4 indicated broad absorption between 400 and 600 nm with two absorption maxima at λ 497 and 503 nm (Figure 1D). SDS-PAGE of the eluents gave characteristic ladder-like bands, with M*r* between 116 and 250 (Figure 1C). These bands were not detected in testis extract from male scallop used as a control (Appendix A). The number and composition of the ladder-like bands varied depending on the storage conditions or freeze-thawing process. We note that fractions without carotenoid absorption (Figure 1C; 3,6–8) also contained the ladder-like bands, suggesting that the apoprotein formed during the handling and separation of the proteins.

### 2.3. Analysis of the Carotenoids

We next analyzed the KBr extract for its carotenoid composition. The pink protein obtained via ammonium sulfate precipitation was analyzed using an HPLC system, and three single peaks, T*_R_* 14.67, 16.03, and 17.44, were mainly detected in a ratio of 65:7:11 (Table 1, Appendix A). These T*_R_* values of the peaks corresponded well with those of pectenolone, pectenoxanthin, and diatoxanthin, respectively (Appendix A). The carotenoid composition of the ovary, 66:10:17, was similar to that of the pink protein, while that of the midgut gland differed greatly (Table 1, Appendix A). The photoabsorption of the primary carotenoid, pectenolone, showed visible light absorption centered around 463 nm, while that of pectenovarin was approximately between 479 and 503 nm, indicating a bathochromic shift—a characteristic feature of carotenoproteins—of about 30 nm on average (Figure 1D,E).

Since this is the first report on the isolation of carotenoprotein from scallops, we named the carotenoprotein “pectenovarin”, as it is a carotenoprotein of scallop ovary mainly possessing pectenolone.

### 2.4. Amino Acid Sequence Analysis

As mentioned above, the purified pink protein was composed of a complex array of proteins. Interestingly, about ten ladder-like bands originally appeared (Figure 1C) and gradually converged into a 95 kDa band after several freeze-thaw processes (Figure 1F, band 4). Five major bands in the SDS-PAGE were cut out and subjected to Edman degradation to determine the N-terminal amino acid sequences. Only the largest 95 kDa band gave amino acid sequences of VQYQK for major and VQRQK for minor N-terminal fragments (Figure 2). We next analyzed the internal amino acid sequences for the five bands (Figure 1F). The gel cut-out was extracted and digested by trypsin, and the resulting peptides were analyzed by high-resolution liquid chromatography/electrospray ionization-quadrupole-time of flight (LC/ESI-Q-TOF) mass spectrometry. Since the following de novo peptide sequencing analysis returned several short sequences having similarity to vitellin proteins, we compared these sequences to a deductive amino acid sequence of vitellogenin from *M. yessoensis* (AGE13945.1) [10,11]. The sequence alignment showed that the inner peptide sequences had approximately 80% similarity to vitellogenin (Figure 2, Appendix A). Of the 76 fragments, 26 fragments completely matched with the vitellogenin sequence; however, the N-terminal sequence found via Edman degradation was not found in the mass-based analysis. An N-terminal-like sequence, VGROL, in the de novo sequencing was found with 14 amino acid residues.

We next compared these peptide fragments to ovorubin, the carotenoprotein from the egg of a freshwater snail, *Pomacea canaliculate*, since this chromoprotein also exhibits pink color [12]. The similarity levels of short fragments obtained from the whole pectenovarin against ovorubin subunits PcOvo1 (AFQ23940.1), PcOvo2 (AFQ23938.1), and PcOvo3 (AFQ23939.1) were 56.9%, 57.3%, and 54.3%, respectively.

## 3. Discussion

In the present study, a new carotenoprotein, pectenovarin, was isolated and characterized, for the first time, from the mature ovary of the Japanese scallop *M. yessoensis* as a pink-colored component. We also analyzed the carotenoid components of the protein. Pectenovarin is a sizeable unstable protein with a molecular size of more than 1000 kDa. SDS-PAGE analysis clearly indicated that pectenovarin is composed of subunits with different molecular weights; however, the unusual behavior of these proteins, especially the observation that the bands converge toward lower molecular bands, requires further investigation.

We found that pectenovarin contained three carotenoids—pectenolone, pectenoxanthin, and diatoxanthin—in about a 65:7:11 ratio under AS precipitation, similar to their ratio in the ovary. These carotenoids correspond to the dominant three of the five carotenoids of scallop ovary [5], which suggests that these carotenoids mostly accumulate in the ovary as pectenovarin in a mature stage.

We also found that the association of carotenoid to the protein resulted in a bathochromic shift of more than 20 nm, leading the compound to be categorized as a “true carotenoprotein” [13].

De novo peptide sequencing of the whole pectenovarin revealed that the internal sequences correspond well to those of vitellogenin from *M. yessoensis* ovary [10,11], with about 80% similarity. Apo-pectenovarin is likely to occur from vitellogenin (Vg) or Vg-like protein. In the noble scallop *Chlamys nobilis*, vitellogenin gene expression was shown to be correlated to the total carotenoid contents [14], which is likely to be directly underpinned by our results.

A short N-terminal amino acid sequence of a pectenovarin subunit was found just adjacent to the putative signal peptide of Vg [10,11]. Notably, the N-terminal amino acid sequence VQRQK, however, was found as a minor component corresponding to VQRQK of residues 17–21 in the Vg sequence, while the major N-terminal sequence, VQYQK, had a mutation at residue 3. In the de novo LC-MS/MS sequencing, no peptide fragment corresponding to the N-terminal residues was found in the present study. Although this may be due to experimental limitations, it is worthy to note that some vitellogenins of avian, amphibian, and fish origin are reportedly products of plural gene coding [15,16,17], and pectenovarin could be produced by the processing of plural vitellogenins. Although egg carotenoproteins are widely distributed among gastropods, bivalves, and crustaceans, investigations at the molecular level are limited. The fact that pectenovarin was found to have about 60% similarity with ovorubin, a pink egg carotenoprotein only characterized from the apple snail *P. canaliculata*, suggests that egg carotenoproteins originated generally from vitellogenins and have evolved to have certain functions to adapt to different environments [18,19]. Interestingly, the Japanese scallop and apple snail differ largely in their reproduction ecology, as the former is a marine broadcast spawner, but the latter is a terrestrial snail depositing its eggmass outside the water. This explains reports that the apple snail carotenoprotein is toxic, as the eggs need to be protected from potential predators [20]. Pectenovarin, a component of edible scallop ovary, may have functions different from those of ovorubin to protect pelagic eggs during the early stage of development. The function might be related to anti-oxidation and photo protection considering the functions of the carotenoids contained in the protein. Notably, pectenolone, a dominant carotenoid of pectenovarin, was reported to inhibit lipid peroxidation, and its ability was almost as potent as astaxanthin, known to possess high-antioxidant activity [6]. It is likely that the protein functions as a delivery and storage system of unstable functional small molecules in the ovary.

Structural studies of carotenoproteins at the molecular level were found only for crustacyanin of crustaceans, orange carotenoprotein of cyanobacteria, and, to some extent, ovorubin. The lack of gene and amino acid sequences and structural investigations of carotenoprotein hampers comprehensible discussion as to the physiological role, mode of association with carotenoids, and evolutional aspects of its presence. The present results showing the extremely complex nature of egg carotenoprotein suggest future directions using genetic experiments to deduce the whole amino acid sequences of the subunits and cryo-electron microscopy to reveal its structure. These studies are now in progress and will be reported elsewhere.

## 4. Materials and Methods

### 4.1. Materials

Live scallops (*Mizuhopecten yessoensis*) were purchased at a local grocery store and stored at −30 °C until use. Chemical reagents were purchased from Wako pure chemicals unless otherwise noted.

### 4.2. Extraction of the Colored Aqueous Components from the Scallop Ovary

Approximately 50 g of the scallop ovary was homogenized with 150 mL of 50 mM Tris-HCl pH 8.0, and then the mixture was centrifuged at 9000 rpm at 4 °C for 30 min. The collected pink-colored supernatant was tested for coloristic stability against the following acidic buffers containing 20 mM NaCl: a buffer of pH 3.5 made of 50 mM citric acid–NaOH, and another buffer of pH 4.0 made of 50 mM AcOH–AcONa. Volumes of 0.5 mL of the specimen and each buffer were mixed, then centrifuged at 12,000 rpm at 4 °C for 15 min.

Another approximately 60 g of ovary specimen was homogenized with 50 mM Tris-HCl (pH 7.2) containing 20 mM NaCl as a neutral condition. The homogenate was centrifuged at 9000 rpm at 4 °C for 30 min, and then the pink-colored precipitate was suspended using the same buffer but at pH 8.0. The above procedure was repeated with buffers containing 150 mM NaCl.

For the optimum extraction buffer, KBr was added to the buffer (50 mM Tris-HCl, pH 8.0) up to 1.12 g/mL so that pink-colored solution was obtained efficiently from the tissue. Twenty grams of scallop ovary were homogenized with the above buffer, and the mixture was centrifuged at 9000 rpm at 4 °C for 30 min; then, the deep-pink-colored supernatant was collected.

### 4.3. Isolation of Pectenovarin

#### 4.3.1. Extraction of the Colored Aqueous Component

The colored aqueous component was extracted anew from the ovary according to the above procedure. Specifically, 105.8 g of the specimen was homogenized with 50 mM Tris-HCl pH 8.0/150 mM NaCl containing KBr (*d* = 1.21). The centrifuged supernatant (3 mL) was layered on 10 mL of 45% KBr (saturated) and centrifuged at 15,000 rpm at 4 °C for 3 h. Then, 300 μL of red supernatant was mixed with 600 μL of saturated KBr in water, which was incubated on ice overnight. Then, the sample was centrifuged (15,000 rpm, 16 °C, 3 h) and repeatedly cooled on ice to remove white floating material. Finally, the extract was concentrated using an Amicon Ultra centrifugal filter (100 kDa, Merck KGaA, Darmstadt, Germany, 5000 rpm at 4 °C for 30 min.). The sample solution was filtered using a 0.45 μm syringe filter.

#### 4.3.2. Gel Fractionation Chromatography

The above extract was separated via high-pressure liquid chromatography (HPLC) using a BioSec-5 gel filtration column (300Å, 4.6 × 30 mm, Agilent technology Japan, Tokyo) in several loads and eluted by 50 mM Tris-HCl pH 8.0/150 mM NaCl at 0.35 mL/min. The eluents were monitored using a multiwavelength detector (MD-910, JASCO Corporation, Tokyo, Japan). Portions of the eluents were analyzed by 8% acrylamide gel SDS-PAGE [21], and then the pale-pink-colored eluents were combined and concentrated by 100 kDa cut-off centrifugal ultrafiltration. The protein quantity of this concentrated specimen was confirmed using a spectrophotometer (NanoDrop2000c, Thermo Fisher Scientific, DE, USA) to 1.667 mg/mL at 20 times dilution.

### 4.4. Analysis of Carotenoids

#### 4.4.1. Precipitation of Analytical Materials

Ammonium sulfate precipitations of the pink aqueous extracts were prepared to estimate the carotenoid composition of pectenovarin. First, three ovary specimens (23.3, 16.1, and 17.5 g) were homogenized separately with 80 mL of 0.05 M Tris-HCl pH 8.0/150 mM NaCl containing KBr (*d* = 1.21). After centrifugation (9000 rpm, 4 °C, 30 min), ammonium sulfate was added up to 10% into the reddish supernatants and stirred on ice to dissolve completely. Then, the mixture was cooled on ice for 30 min and centrifuged for 15 min at 12,000 rpm at 4 °C. The concentration of AS was increased until colorless supernatants were observed. The addition of ammonium sulfate up to 25% finally brought the reddish-pink color into the pellet completely. The freeze-dried pellets (13.5, 14.3, and 11.8 mg) were then adjusted to 100 mg/mL with AcCN/MeOH/water at a ratio of 50:50:1. In order to compare carotenoid compositions between the pectenovarin and tissues of the scallop, freeze-dried ovary and midgut gland were also analyzed. First, the freeze-dried specimens of the ovary (131.8, 136.5, and 133.6 mg) and midgut gland (136.8, 133.5, and 133.7 mg) were moistened with water and then extracted three times with 1 mL of acetone. The extracts were partitioned between hexane and water, which separated them into an orange-colored upper layer and a cloudy aqueous layer. The upper layer was collected and evaporated while protecting it from light and then adjusted to 10 mg/mL using the above solvent system.

#### 4.4.2. HPLC Analysis

The above sample (50 μL) was loaded into an HPLC column (Capcell Pack C18-UG80, 4.6 × 30 mm, OSAKA SODA CO., LTD., Osaka, Japan) and then eluted using AcCN/MeOH/water at a ratio of 50:50:1 at 0.35 mL/min. The eluents were monitored using a photodiode array detector (L-7455, Hitachi Ltd., Tokyo, Japan). In order to compare the carotenoid compositions between the AS precipitate, ovary, and midgut gland, the ratios of carotenoid peak area on these chromatograms at 463 nm, which is the maximum absorption of pectenolone, were calculated.

#### 4.4.3. Identification of Carotenoids Using LC-MS/MS

LC-MS/MS was performed in positive mode on a Q-TOF ESI MS/MS (SCIEX TripleTOF, Sciex, MA, USA) equipped with HPLC (Shimadzu Co., Kyoto, Japan) using a BEH C8 column (2.1 × 150, S 2.5 um, Waters, MA, USA) with a gradient of solvent A: 0.1% formate/10 mM ammonium formate containing water, B: 0.1% formate/10 mM ammonium formate containing MeOH/2-propanol (85:15, *v*/*v*). The flow rate was set to 0.3 mL/min and the time program was B concentration 75%, 0–2 min; 75%–99%, 2–18 min; 99%, 18–24 min; 99–75%, 24–25 min; 30 min stop. The column oven temperature was 50 degrees, and 5 μL of sample was injected.

### 4.5. N-terminal Amino Acid Sequence Analysis

Polypeptides on the 6% acrylamide gel after running of SDS-PAGE were transferred to a PVDF membrane using an iBlot Gel Transfer System (Thermo Fisher Scientific Inc., MA, USA). The polypeptides on the PVDF membrane were dyed using acid red 112 (Tokyo Chemical Industry Co., Tokyo, Japan) and then cut out from the membrane and loaded onto a Procise 492HT protein sequencer (PerkinElmer, Inc., MA, USA).

### 4.6. Internal Amino Acid Sequence Analysis

A 3 μL pectenovarin specimen, with protein concentration 14 μg/μL, was loaded onto 6% polyacrylamide gel for SDS-PAGE, and the separated five bands were respectively cut out and kept in 1.5 mL microtubes. These specimens were reduced using dithiothreitol and then alkylated using iodoacetamide and in-gel. The gel was immersed in 50 mM ammonium hydrogen carbonate; then, the protein was digested in-gel using TPCK trypsin with a molar ratio of 1:50 and incubated for 20 h at 37 °C. The enzyme digests were extracted from the gel using 5% formic acid/50% acetonitrile. The solvents were removed and then dissolved with 80 μL of water and analyzed by LC-MS using a C18 column (Vydac PROTEIN and PEPTIDE C18). The sequences of fragments were extracted using DeNovoGUI 1.16.4 and analyzed and compared using the FASTS program, which finds regions of similarity among proteins with unordered peptides against target sequences [22].

## 5. Conclusions

In the present study, we report the isolation and properties of pink carotenoprotein from the ovary of edible Japanese Scallop, *Mizuhopecten yessoensis*. The carotenoprotein, named pectenovarin, is a large and complex protein composed of more than ten subunits. Extracting the protein from the ovary was a challenge due to the poor stability of the molecule. The extractability of the protein varied from individual to individual, even when using live specimens. The best separation was finally achieved by applying the extract directly to a gel filtration HPLC column. Our success in obtaining reasonably pure protein paved the way for studying the structure of this intriguing molecule. The partial sequence obtained here showed the close relation of the carotenoprotein to vitellogenin. It seems, however, that the protein was highly processed from the vitellogenin of the scallop. Pectenovarin complexes with pectenolone and a few other minor related carotenoids. The mode of complexation is interesting as all other marine carotenoproteins complex solely with astaxanthin. Together, the present results illustrate the interesting utilization of carotenoids in the scallop, in that food-derived carotenoids are converted to pectenolone by the animal and perhaps delivered to the ovary by the formation of a complex with apo-pectenovarin; the complexed protein, pectenovarin, is then stored in the ovary. The structure and behavior of pectenovarin were more complex than expected, but further studies on the unique utilization of carotenoids would provide interesting insights into the physiological and evolutionary aspects of scallops.

## Figures and Tables

**Figure 1 molecules-25-03042-f001:**
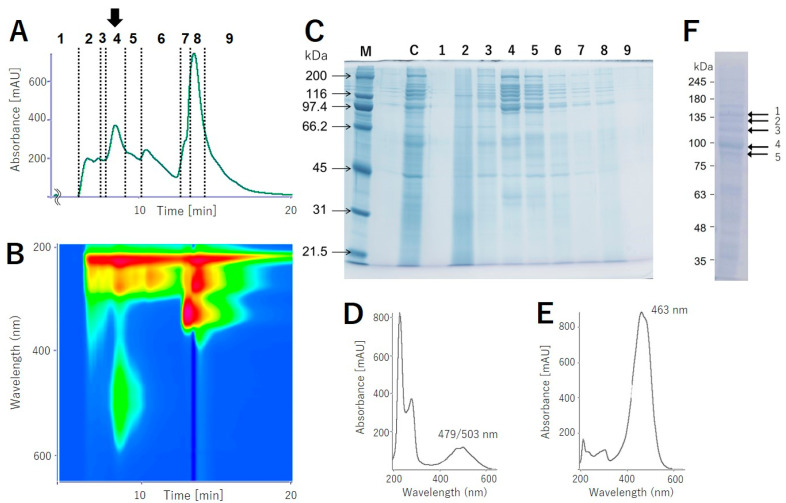
(**A**) The chromatogram at 280 nm and (**B**) HPLC absorbance spectrum from pectenovarin purification using a photodiode array detector (PDA). (**C**) SDS-PAGE of the HPLC fractions. M: size marker; C: the crude extract of the scallop ovary; 1–9: the fraction numbers corresponding to Figure 1A. (**D**) Absorbance spectrum of pectenovarin at T*_R_* = 8.459 min and (**E**) pectenolone in acetonitrile/methanol/water at 50:50:1. The peak maxima of the carotenoid chromophore of pectenovarin are at 479 and 503 nm, while that of pectenolone is at 463 nm. (**F**) The ladder-like bands originally appeared and gradually converged at 95 kDa (band 4) after several freeze-thaw processes.

**Figure 2 molecules-25-03042-f002:**
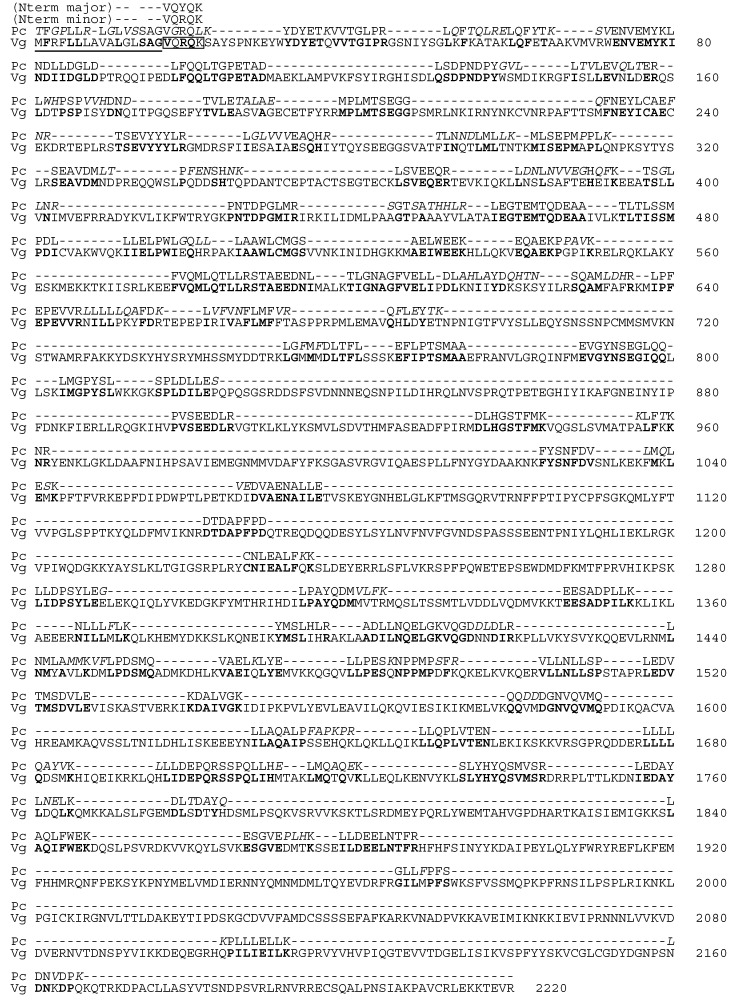
Sequence comparison between the fragments of pectenovarin and vitellogenin. Peptide fragments obtained by de novo LC-MS/MS analysis of pectenovarin were aligned to the sequence of vitellogenin from *M. yessoensis*. Boldface letters indicate exact-match residues (note that L and I were regarded as the same residue). Italic letters are unmatched residues. The boxed sequence corresponds to the N-terminal sequence shown by Edman degradation of native pectenovarin (Nterm major or minor). The underline indicates the predicted cleavable signal peptide (16 residues) [10,11]. Pc: pectenovarin (MS sequencing fragments), Vg: vitellogenin of *M. yessoensis* (AGE13945.1).

**Table 1 molecules-25-03042-t001:** Carotenoid compositions of the organs of scallop *M. yessoensis.*

Carotenoid	% (Peak Areas at 463 nm) ^1^			
	AS ^3^ Precipitation	Ovary	Midgut Gland *	Matsuno et al.
Peak 1 ^2^	3.8	3.0	44.8	-
Pectenolone	64.7	65.8	3.8	73.0
Pectenoxanthin	7.2	10.2	5.1	9.0
Diatoxanthin	11.2	16.5	2.7	13.0
Peak 5 ^2^	3.4	0.1	-	-
Other	16.9	7.5	88.4	4.0

^1^ Analyzed by PDA. ^2^ Structures are not assigned. ^3^ AS, Ammonium sulfate, * These analyses were performed in triplicate; however, data from one sample was used for the midgut gland because large individual differences and inseparable peaks were found.

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
