# Peer review of "Pectenovarin, A New Ovarian Carotenoprotein from Japanese Scallop Mizuhopecten yessoensis"

_molecules, 2020, doi:10.3390/molecules25133042_

Round 1

Reviewer 1 Report

The article by Matsunaga, Ikeda and Sakai titled "Pectenovarin, a New Ovarian Carotenoprotein from Japanese Scallop Mizuhopecten yessoensis" describes the isolation of a new protein closely related to vitellogenins and responsible to the pink colour of scallop ovary.

The article is clear and neat. It reminds me of classical biochemical papers and I enjoyed the reading.

I have two comments:

Authors use term pectenovarin in Fig. 1, however, it is revealed for the first time in subsequent section 2.3. Such word usage could mislead the reader. I would ask to consider either disclose the term in section 2.2 (since it has "pectenovarin" in heading) or change the Fig. 1 legend to avoid the use of the undisclosed term.

Line 180 says that pectenovarin is not toxic. This statement seems not to be grounded in experimental data. I understand that the whole ovary is probably not toxic to human. However, the isolated protein toxicity has not been studied and only could be providently assumed.

Author Response

Thank you for your comment and the suggestion. As for the first suggestion, we agree with you and we inserted the description on “pectenovarin” after “In order to isolate colored protein” of the first sentence of 2.2 (line 74).

All the reviewers came up with similar suggestions. According to the suggestion of them, we changed a line from "Pectenovarin, a non-toxic edible protein, may have functions different to those of ovorubin to protect pelagic eggs during the early stage of development." to “Pectenovarin, a component of edible scallop ovary, may have functions different from those of ovorubin to protect pelagic eggs during the early stage of development. The function might be related to anti-oxidation and photoprotection, considering the functions of the carotenoids contained in the protein. Of note, pectenolone, a dominant carotenoid of pectenovarin, was reported to inhibit lipid peroxidation, and its ability was almost as potent as astaxanthin that is known to possess high antioxidant activity [6]. It is therefore likely that the protein functions as delivery and storage system of an unstable functional small molecule in the ovary.” (line183-189)

Reviewer 2 Report

In the study “Pectenovarin, a new ovarian carotenoprotein from Japanese scallop Mizuhopecten yessoensis” Satoko Matsunaga, Hiroki Ikeda and Ryuichi Sakai describe the isolation of new carotenoprotein termed pectenovarin from the ovary of Japanese scallop. The authors using a range of separation and analytical techniques isolated and purified the protein, determined its amino acid sequence and the carotenoids complexed with the protein. The amino acid sequence of the new protein showed a high similarity with vitellogenin, the precursor of egg yolk protein.

This is a very well thought out and systematically done study to characterize a new carotenoprotein from Japanese scallop ovary.

Comment: A little discussion on the potential biological functions of the protein is suggested.

Author Response

Thank you for your comment and suggestion. We expanded the discussion regarding the possible function of pectenvarin with one new reference. Thus, we changed line183-189 as “Pectenovarin, a component of edible scallop ovary, may have functions different from those of ovorubin to protect pelagic eggs during the early stage of development. The function might be related to anti-oxidation and photoprotection, considering the functions of the carotenoids contained in the protein. Of note, pectenolone, a dominant carotenoid of pectenovarin, was reported to inhibit lipid peroxidation, and its ability was almost as potent as astaxanthin that is known to possess high antioxidant activity [6]. It is therefore likely that the protein functions as delivery and storage system of an unstable functional small molecule in the ovary.”

Reviewer 3 Report

The manuscript is quite interesting. The authors characterize a new and complex protein.
They also performed an interesting chemical analysis and identified different carotenoids. However, they supplied no information on their physiological significance even from literature. Therefore, they should improve the discussion, adding to the manuscript all of the information they find.
They should significantly improve Figure 1C.

Author Response

Thank you for your comment and suggestion. According to you and other reviewer’s comments, we added lines 185-189 to discuss possible functions of pectenovarin as follows with one additional reference. “The function might be related to anti-oxidation and photoprotection considering the functions of the carotenoids contained in the protein. Of note, pectenolone, a dominant carotenoid of pectenovarin, was reported to inhibit lipid peroxidation, and its ability was almost as potent as astaxanthin that is known to possess high antioxidant activity [6]. It is therefore likely that the protein functions as delivery and storage system of an unstable functional small molecule in the ovary.”

As you pointed out, the legend for figure 1C was incomplete, and thus readers might be difficult to follow. Now new legend was added as follows: “M: size marker; C: the crude extract of the scallop ovary; 1-9: the fraction numbers corresponding to Figure1A.” (line92-94).